# Bidirectional Association between Metabolic Control in Type-2 Diabetes Mellitus and Periodontitis Inflammatory Burden: A Cross-Sectional Study in an Italian Population

**DOI:** 10.3390/jcm10081787

**Published:** 2021-04-20

**Authors:** Federica Romano, Stefano Perotto, Sara Elamin Osman Mohamed, Sara Bernardi, Marta Giraudi, Paola Caropreso, Giulio Mengozzi, Giacomo Baima, Filippo Citterio, Giovanni Nicolao Berta, Marilena Durazzo, Gabriella Gruden, Mario Aimetti

**Affiliations:** 1Department of Surgical Sciences, C.I.R. Dental School, Section of Periodontology, University of Turin, 10126 Turin, Italy; federica.romano@unito.it (F.R.); sara.amosm@gmail.com (S.E.O.M.); marta.giraudi@gmail.com (M.G.); giacomo.baima@unito.it (G.B.); filippo.citterio@unito.it (F.C.); 2Postgraduate Program in Periodontology, C.I.R. Dental School, University of Turin, 10126 Turin, Italy; stefanoperotto@libero.it; 3Department of Medical Sciences, University of Turin, 10126 Turin, Italy; bernardi.sara98@gmail.com (S.B.); marilena.durazzo@unito.it (M.D.); gabriella.gruden@unito.it (G.G.); 4Clinical Biochemistry Laboratory, Department of Laboratory Medicine, 10126 Turin, Italy; pcaropreso@cittadellasalute.to.it (P.C.); giulio.mengozzi@unito.it (G.M.); 5Department of Clinical and Biological Sciences, University of Turin, 10043 Orbassano, Italy

**Keywords:** glycated hemoglobin, inflammation, periodontitis, periodontal inflamed surface area (PISA), type-2 diabetes mellitus

## Abstract

This study assessed the periodontal conditions of type 2 diabetes (T2DM) patients attending an Outpatient Center in North Italy and explored the associations between metabolic control and periodontitis. Periodontal health of 104 T2DM patients (61 men and 43 women, mean age of 65.3 ± 10.1 years) was assessed according to CDC/AAP periodontitis case definitions and Periodontal Inflamed Surface Area (PISA) Index. Data on sociodemographic factors, lifestyle behaviors, laboratory tests, and glycated hemoglobin (HbA1c) levels were collected by interview and medical records. Poor glycemic control (HbA1c ≥ 7%), family history of T2DM, and C-reactive protein levels were predictors of severe periodontitis. An increase in HbA1c of 1% was associated with a rise in PISA of 89.6 mm^2^. On the other hand, predictors of poor glycemic control were severe periodontitis, waist circumference, unbalanced diet, and sedentary lifestyle. A rise in PISA of 10 mm^2^ increased the odds of having HbA1c ≥ 7% by 2%. There is a strong bidirectional connection between periodontitis and poor glycemic control. The inflammatory burden posed by periodontitis represents the strongest predictor of poor glycemic control.

## 1. Introduction

The prevalence of diabetes mellitus (DM) is increasing worldwide to epidemic proportions: 415 million people suffer from DM and the number is expected to rise to 642 millions by 2040 [1]. DM has two major types: Type 1 (T1DM), characterized by failure to produce insulin, and type 2 (T2DM), in which both insulin resistance and relative insulin deficiency occur. T2DM is the most prevalent form of the disease, accounting for more than 90% of diabetic patients [2].

DM presents a serious challenge to the healthcare system since its complications are the leading causes of morbidity and mortality. According to the World Health Organization, it will be the seventh major cause of death in 2030 [3]. Systemic subclinical inflammation has been proposed as the underlying biological mechanisms of its chronic complications, such as micro vascular and nerve damage [4], with evidence of a strong association between levels of hemoglobin A1c (HbA1c) and risk of complications [5].

Periodontitis is recognized as the sixth most common complication for both DM forms even if most of the studies are related to T2DM [6]. Periodontitis is an infectious disease with a chronic inflammatory response to periodontal pathogens in dental biofilm that leads to the irreversible destruction of the tooth-supporting tissues and eventually to tooth loss. There is consistent and robust evidence supporting the existence of a relationship between T2DM and periodontitis with a dual directionality [7,8,9]. T2DM enhances the risk for periodontitis initiation and progression, and periodontal inflammation affects both glycemic control and the risk to develop chronic T2DM complications [10,11,12]. A hyperglycemic status leads to a dysregulated inflammatory response involving immune activity, neutrophil functioning, and cytokine pattern, promoting connective-tissue damage [13,14]. On the other hand, the dissemination of periodontal pathogens and their metabolic products in the bloodstream circulation, results in increased serum levels of inflammatory mediators that can deteriorate blood glucose control via acute-phase (i.e., C-reactive protein, CRP) and neutrophil oxidative response [15,16].

This systemic inflammatory burden has proven to increase with the extent and severity of periodontitis. Conventionally, the overall amount of destruction of tooth-supporting tissues is measured in terms of clinical attachment level (CAL) and probing depth (PD). However, these clinical parameters assess the cumulative effects of periodontal tissue breakdown, but do not measure the amount of inflamed and ulcerated epithelium within the periodontal pocket [17]. The Periodontal Inflamed Surface Area (PISA) Index has been introduced to quantify the amount of bleeding pocket epithelium and it is expected to reflect the inflammatory burden presented by periodontitis [18,19]. PISA values tend to increase consistently as periodontal status worsens, even if they show high variability in studies conducted in different populations [20,21,22,23].

Although the association between periodontitis and T2DM has been widely demonstrated, the strength of the relationship seems to differ geographically (based on genetic and lifestyle differences among ethnic groups) [24,25]. Furthermore, little is known on the behavior of PISA in T2DM. A recent study has reported a dose–response relationship between PISA values and HbA1c levels [20].

The aims of the present cross-sectional study were to determine the periodontal health status of T2DM patients attending an Outpatient Diabetes Center in North Italy [26] and to assess the association between glycemic control and periodontitis, as measured clinically and with the PISA Index.

## 2. Materials and Methods

### 2.1. Study Design

This cross-sectional study was conducted in accordance with the Helsinki Declaration and approved by the Institutional Ethical Committee of the “AOU Città della Salute e della Scienza”, Turin, Italy (No. 0027219, 14 March 2018). Informed consent was obtained from each patient before the study. All participants signed an informed consent to undergo physical and periodontal examination. The study complied with the STROBE guidelines.

Patients with an established diagnosis of T2DM according to World Health Organization criteria [27] were consecutively recruited from among those who came for regular check-ups at the Outpatient Diabetes Center, Turin (Italy) from March 2018 to July 2019.

The following inclusion criteria were considered: (i) at least 40 years of age; (ii) having at least 8 teeth; (iii) availability of measurements of routine diabetes laboratory tests made in the 6 months before enrollment. Exclusion criteria were: (i) T1DM; (ii) intake of drugs known to affect gingival tissues, use of antibiotics, steroidal, and/or non-steroidal anti-inflammatory drugs 3 weeks prior to the visit; (iii) periodontal therapy in the past 6 months; (iv) pregnancy or lactation; and (v) diagnosis of following pathologies: cancer, human immunodeficiency virus/AIDS, chronic infections, liver/kidney failure excluding diabetic nephropathy, chronic obstructive pulmonary disease with acute episodes and/or requiring the use of steroidal inhalator.

### 2.2. Data Collection

Participants were required to complete a questionnaire to obtain information on socio-demographic characteristics (gender, age, ethnicity, education), general health behavior (leisure-time physical activity level, daily smoking and dietary habits, alcohol consumption), and oral hygiene behavior (toothbrush frequency, use of interdental devices, and frequency of professional oral hygiene sessions).

Data on medical history, parental history of T2DM, T2DM onset and duration, cardiovascular risk factors, chronic T2DM complications, current medications and treatment for T2DM, as well as results of laboratory tests performed in the last diabetic visit (HbA1c level, lipid profile, urine analysis, creatinine, high-sensitivity-CRP (hs-CRP)) were collected.

Two masked Diabetes Specialists reviewed the medical history of the participants and conducted a physical examination including blood pressure levels (average of three blood pressure measurements within 3 min), anthropometric measurements (weight, height, and waist circumference (WC)), palpation, and auscultation. The body mass index (BMI) was calculated as weight/height squared (kg/m^2^).

Subsequently, a single dentist conducted a periodontal examination. To ensure inter and intra-examiner reproducibility, measurements of periodontal parameters were repeated in 20% of the sample and compared with those recorded by a gold-standard examiner. The k coefficients (within 1 mm) between examiners ranged from 0.79 to 0.93 in the evaluation of PD and from 0.81 to 0.89 in the evaluation of gingival recession (Rec). The intra-examiner concordance rates for repeated measurements were 0.89 to 0.95 for PD and 0.82 to 0.91 for Rec.

Full-mouth PD, Rec, and CAL were recorded by means of a periodontal probe with 1-mm markings (PCP-UNC 15, Hu-Friedy, Chicago, IL, USA) at six sites per tooth, excluding third molars. The total percentages of sites exhibiting bacterial plaque or bleeding on probing (BoP) were expressed as full mouth plaque score (FMPS) and full mouth-bleeding score (FMBS), respectively. The number of missing teeth was also recorded.

The presence of periodontitis was defined according to the criteria proposed by Centers for Disease Control and Prevention/American Academy of Periodontology (CDC/AAP) for epidemiologic surveys [28,29]. Therefore, moderate periodontitis was defined as at least 2 interproximal sites with attachment loss ≥4 mm (not on the same tooth) or at least 2 interproximal sites with PD ≥ 5 mm, also not on the same tooth. The presence of at least 2 interproximal sites with attachment loss ≥6 mm (not on the same tooth) and at least 1 interproximal site with PD ≥ 5 mm indicated severe periodontitis. If neither moderate nor severe periodontitis applied, no/mild periodontitis was recorded. Additionally, a recently introduced measure of periodontitis severity, the PISA, was calculated as previously described in the literature [18,19]. It quantifies the amount of bleeding epithelium in mm^2^ around individual tooth. The sum of all individual PISAs corresponds to the full-mouth PISA value in mm^2^ of each participant and reflects the inflammatory burden posed by periodontitis.

### 2.3. Statistical Analysis

All data analyses were performed using SPSS software, version 24.0 for MAC (Chicago, IL, USA). Frequency distributions were determined, and descriptive statistics were calculated as means, standard deviations, and ranges. Participants’ tobacco use, adherence to balanced diet, and alcohol consumption were classified as dichotomous variables (yes/no). Education was dichotomized according to the years spent in school, considering a cut-off value of 8 years (lower/high school diploma, university bachelor’s degree or higher).

Categorical grouping variables included periodontal status (no/mild periodontitis, moderate periodontitis, severe periodontitis) and glycemic control (good: HbA1c < 7%; poor: HbA1c ≥ 7%) [30]. Comparisons between groups were performed with chi-square test for categorical variables, unpaired *t*-test or one-way analysis of variance for normally distributed quantitative variables, and with Mann–Whitney *U*-test or Kruskal–Wallis test for non-normally distributed quantitative variables. Post-hoc tests (Scheffé test and Dunn-test with Bonferroni correction) were used for multiple comparisons.

Because severity of periodontitis was operationalized as both a dichotomous variable (absence or presence) according to the CDC/AAP clinical criteria and a continuous variable (full-mouth PISA value), the associations between severity of periodontitis (outcome variable) and glycemic control (primary explanatory variable) was assessed using logistic and linear regression techniques, respectively. Two sets (each containing two models) of multiple regression analyses were fitted separately to each of the outcomes for a total of four models. In each set, glycemic control was entered as (1) dichotomous variable (poor vs. good) and (2) continuous variable (HbA1c levels). In addition, evaluated covariates included hs-CRP (mg/L), presence of chronic diabetes complications (dichotomous) and family history of T2DM (dichotomous).

The predictors of uncontrolled T2DM (outcome variable) were explored by logistic regression analysis (models 5 and 6) including severity of periodontal damage (as dichotomous or continuous variable), WC, leisure-time physical activity, and balanced diet based on previous literature demonstrating the effect of lifestyle factors on T2DM risk and metabolic control [31]. In the logistic regression analysis, odds ratios (ORs) and 95% confidence intervals (CIs) were calculated. The level of significance was set at 0.05.

## 3. Results

The study flow-chart is outlined in Figure 1. A total of 104 T2DM patients, 61 men and 43 women, with a mean age of 65.3 ± 10.1 years were consecutively recruited into this study. All participants were Caucasian. Poor glycemic control (HbA1c ≥ 7%) was detected in 63.5% of T2DM patients.

T2DM patients used oral medications, insulin, or a combination of both to treat and control diabetes. Metformin was the oral medication most often prescribed (77.9%), followed by a new generation of DM drugs (59.6%) and other first-generation agents (20.2%). Insulin therapy was prescribed to 54.8% of the patients.

Table 1 summarizes sociodemographic, lifestyle, and periodontal characteristics stratified by glycemic control. Physical activity level, WC, hs-CRP levels, frequency of chronic DM complications, and periodontal status were significantly different between poorly and well-controlled T2DM patients. Patients with poor glycemic control had higher prevalence of severe periodontitis and PISA values than well-controlled T2DM patients.

When we stratified the analysis by periodontal status (Table 2), patients with severe periodontitis showed statistically significant higher frequency of family history of T2DM, higher HbA1c scores, more chronic DM complications, higher hs-CRP and PISA values than moderate, and no/mild periodontitis patients.

In the multiple logistic regression analysis (Table 3, models 1 and 2), hs-CRP was a predictor of severe periodontitis with an OR of about 1.7 (for both models). Poor glycemic control and a rise in HbA1c of 1% increased by 4.6-fold and by 1.6-fold, respectively, the odds for having severe periodontitis.

In the multiple linear regression analysis (Table 3, models 3 and 4), hs-CRP levels were shown to be significantly associated with PISA, displaying similar β coefficients in both models (46.0 vs. 49.9). Patients with poor glycemic control had higher PISA values (ß = 297.4) and an increase in HbA1c of 1% was associated with a rise in PISA of 89.6 mm^2^. Therefore, the maximum expected area of inflamed epithelium is estimated at 627.2 mm^2^ in patients with well-controlled diabetes. A further increase of 412.2 mm^2^ is expected in the patient with the highest HbA1c level.

Family history of T2DM was an additional predictor only for severe periodontitis, while the presence of chronic DM complications was associated with a rise in PISA.

As depicted in Figure 2, patients diagnosed as having severe periodontitis had mean PISA values of 1454.1 mm^2^, corresponding to the area of a square having sides 38.1 mm each. Inside this square the surface area attributed to the impact of HbA1c measured on average 681.0 mm^2^ (equal to that of a square measuring about 26 mm on each side). The hs-CRP levels accounted for a mean increase in PISA of 154.8 mm^2^ (square of 12.4 mm on each side).

In Table 4, the multiple logistic regression analysis (models 5 and 6) showed that poor lifestyle habits were predictors of poor glycemic control. A rise in PISA of 1 mm^2^ increased the odds of having HbA1c ≥ 7% by 2‰, and patients with severe periodontitis were 8.5-times more likely to have uncontrolled diabetes.

## 4. Discussion

This cross-sectional study aimed to assess the periodontal conditions of T2DM adult patients attending an Outpatient Clinic in the North of Italy, and to examine predictors of the association between periodontitis and glycemic control. In spite of the large number of studies investigating this latter aspect, only a few of them focused on the impact of the inflammatory burden of periodontitis assessed using the PISA Index and none referred to the CDC/AAP periodontitis case definition [18,19,20]. The present study used a full-mouth examination protocol and applied this classification system [28,29]. These criteria have been recommended when investigating systemic periodontal linkages [32].

Previous investigations on T2DM reported largely variable percentages of periodontitis, ranging between 13.6% and 97.7%, as a result of differences in both ethnic background and measures of periodontitis severity [25,33,34,35,36,37]. In an epidemiologic survey on our Caucasian population, 52% of all adults over 60 of age, irrespective of their diabetes status, suffered from severe periodontitis based on CDC/AAP algorithms [38]., Futhermore, the percentage of periodontitis was 91% with 63.4% of the severe form, reflecting an urgent need for treatment and preventive oral care programs.

Although the two-way relationship between periodontitis and T2DM has been long established, multivariable modelling procedures have rarely been applied to explore such an association from both perspectives. We used two methods to operationalize severe periodontitis: the first method relied on clinical examination of PD and CAL based on CDC/AAP case definition, and the second one measured the surface area of bleeding periodontal epithelium according to the PISA Index [18,19]. This parameter, recently introduced in periodontal medicine research, is based on mathematical algorithms difficult to apply during the routine clinical practice but it could be a valid method to numerically represent the active inflammatory status of periodontium.

When periodontitis was entered as a dependent variable in the multivariate analysis, family history of T2DM, glycemic control, and serum hs-CRP levels were found to be significantly associated with an increased odds for having poorer periodontal health. These findings were irrespective of the definitions used for periodontitis and corroborate with those documented in the literature [9,39]. Familiarity was reported by approximately 70% of patients with severe periodontitis compared with 20% and 10% of individuals with moderate and no/mild periodontitis, respectively.

Individuals suffering from uncontrolled T2DM were at least four times more likely to have severe periodontitis than those with better-controlled diabetes. About 63.5% of T2DM patients had poor glycemic control, with a trend to suggest higher HbA1c levels with increasing periodontitis severity. This percentage is high compared with data previously published in the literature, reporting that one-third to half of diabetics having an HbA1c level ≥ 7% [36,40,41]. Our study population derives from a specialized diabetic center where severe cases are referred to. Patients had a mean duration of T2DM of 14.2 ± 10.7 years, long enough for chronic complications of the disease to appear.

Interestingly, the strength of the association of HbA1c levels with severe periodontitis was comparable to that of serum hs-CRP, a marker of systemic inflammation [42]. A one-unit rise in both serum HbA1c and hs-CRP could increase the odds for having severe periodontitis by approximately 60%. Serum reactive oxygen species, interleukin (IL)-1, IL-6, tumor necrosis factor (TNF)-α, and CRP have been found to be elevated in the bloodstream of patients with established T2DM and may play an important role in tissue breakdown in periodontitis [43]. This suggests that as the severity and duration of chronic hyperglycemia increase, the periodontal inflammatory response is also expected to rise.

When considering the PISA Index as dependent variable, the mean surface area of ulcerated epithelium was estimated 288.33 mm^2^ higher in uncontrolled than that in well-controlled T2DM, and an increase in HbA1c of 1% was associated with a rise in PISA of 89.6 mm^2^. The PISA increased by 46 mm^2^ for every one-unit increase in hs-CRP concentration. To date, only one study reported data on this outcome and found PISA as a predictor of periodontitis severity with an expected increase of 275.29 mm^2^ in T2DM compared to non-diabetics [44].

On the other hand, we explored the relationship between glycemic control, dependent variable, and periodontitis. Predictors of poor glycemic control were periodontitis, WC, unbalanced diet, and sedentary lifestyle habits. Patients affected by severe periodontitis were eight times more likely to have uncontrolled T2DM than those with moderate or no periodontitis. Odds were increased by 2% for a ×10 mm^2^ PISA raise. Only two studies investigated the association between periodontitis and HbA1c with conflicting results [20,34]. Susanto et al. did not find any association in Indonesians with T2DM, but they identified PISA as predictor of HbA1c together with CRP in non-diabetic controls [34]. Conversely, a dose–response relationship was previously reported between PISA and HbA1c in T2DM patients living in Carribean island Curaçao [20].

An interesting finding in this study was the absence of any statistically significant association of plasma lipid profile and overweight/obesity (BMI ≥ 25 kg/m^2^) with severity of periodontitis in both the bivariate and multivariate analysis. It cannot rule out the possibility that the effects of obesity and low HDL levels, that are major components of the metabolic syndrome, may be masked by the hyperglycemic status [45]. The present data support a relationship between inadequate glycemic control and poor lifestyle habits. Although regular exercise improves insulin sensitivity and regulates blood glucose levels, it is often difficult for T2DM patients to maintain regular exercise habits [46,47]. In our study, only one-third of poorly controlled diabetics reported that they exercised regularly.

Biological mechanisms explaining the effect of periodontitis on the metabolic control have been proposed in the literature, but the actual evidence is moderate [8]. Local inflammation can affect systemic health through this ulcerated interface, allowing bacteria and inflammatory mediators to access to the bloodstream [48,49,50,51,52]. The amount of ulcerated epithelial area, according to the study by Leira et al. [21], was between 934 and 3274 mm^2^ in patients with severe periodontitis. These values are higher than those observed in the present severe periodontitis group ranging between 585 and 2732 mm^2^ (mean value 1454 mm^2^). In recent years, the surface area of ulcerated epithelium in contact with the subgingival biofilm, previously estimated to be large as the palm of an adult hand (about 7000 mm^2^) [53], has been resized to values ranging between 800 and 2000 mm^2^ in patients diagnosed with periodontitis [18]. The present data confirm these previous findings.

There is evidence of reduced pancreatic beta-cell function and increased low-grade inflammatory burden in individuals suffering from periodontitis, which in turn affects lipid metabolism and contributes to increased insulin resistance and poor glycemic control [54,55,56,57]. This has relevant implications because chronic inflammation represents the biological linking mechanism between periodontitis, T2DM, and its complications [8,9]. A recent systematic review associated diabetes-related retinopathy, nephropathy, and neuropathic foot ulceration to the severity of the periodontal damage [15]. Consistently, effective non-surgical periodontal treatment has resulted in statistically significant reduction of HbA1c levels of about 0.40% at 3 months in T2DM patients [58].

This study has some limitations. A cause–effect interpretation cannot be determined in a cross-sectional survey limiting the strength of the conclusions. The sample size was also limited owing to the use of restricted enrollment criteria. A notably high number of patients were excluded because they had less than eight remaining teeth. As reported by Susanto et al. [34], the application of the PISA Index requires the presence of at least eight natural teeth. Accordingly, a minimum number of four to six natural teeth is needed when using CDC/AAP algorithms to ensure the measurements necessary to diagnose periodontitis and to minimize its misclassification [59,60]. Therefore, the association between periodontitis and T2DM cannot be generalized to almost and completely edentulous persons even though they may have had a past history of periodontal disease.

Furthermore, in spite of the well-established correlation of smoking habits with both poor glycemic control and periodontitis, the number of smokers was small and unable to be included in the statistical analysis [61,62]. Last, we selected a non-random convenience sample of diabetics from an Outpatient Diabetic Center that probably provided medical care to more severely affected T2DM people than the community-based diabetic population. This may limit the generalizability of the present findings.

## 5. Conclusions

There is a strong association between severe periodontitis and poor glycemic control. The predictors were different when considering the relationship between the two diseases in both directions. The inflammatory burden posed by periodontitis, as measured by the PISA score, represents the strongest predictor of poor glycemic control in T2DM. To the best of our knowledge, this is the first study addressing these clinically relevant aspects using the PISA Index and the CDC/AAP periodontitis case definition. This finding is in line with the prominent role played by poorly controlled T2DM in the progression rate of periodontitis as recently emphasized in the International Workshop on Classification of Periodontal Diseases [30] and by the Joint Workshop of the International Diabetes Federation and the European Federation of Periodontology [8].

## Figures and Tables

**Figure 1 jcm-10-01787-f001:**
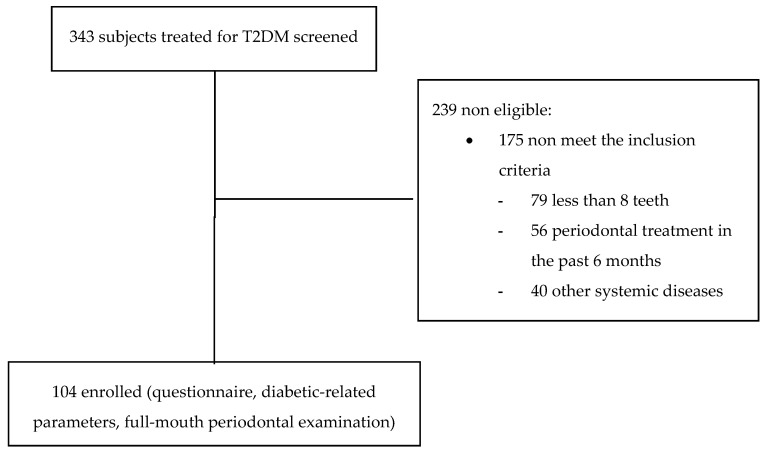
Subject recruitment and participation.

**Figure 2 jcm-10-01787-f002:**
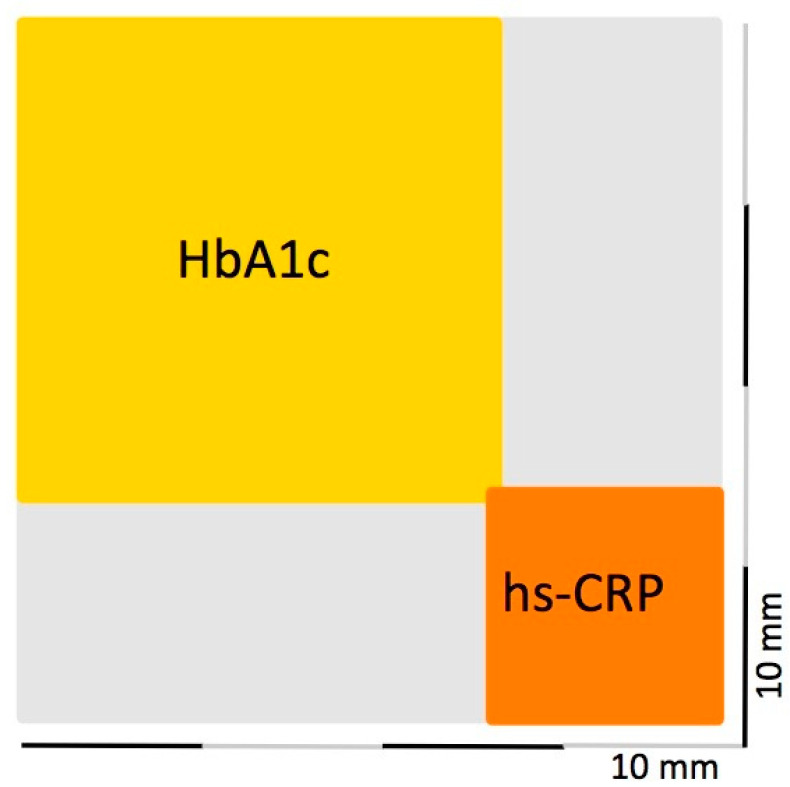
Square in mm^2^ representing the mean PISA values for patients diagnosed as having severe periodontitis and depicting the surface area explained by glycated hemoglobin (HbA1c, yellow) and high-sensitivity C reactive Protein (hs-CRP, orange) levels. The grey area identifies the surface area attributable to other unexplored factors that influence the PISA.

**Table 1 jcm-10-01787-t001:** Sociodemographic and clinical characteristics of T2DM patients (mean ± SD (range) or *n* (%)) according to the glycemic control.

Variable	Poor Glycemic Control (*n*= 66)	Good Glycemic Control (*n* = 38)	Total (*n* = 104)	*p* Value
Age (years)	65.9 ± 9.7 (43–80)	64.4 ± 10.8 (40–78)	65.3 ± 10.1 (40–80)	0.459
Sex				0.076
Males	43 (70.5)	18 (29.5)	61 (58.7)	
Females	23 (53.5)	20 (46.5)	43 (41.3)	
Education level				0.796
<high school	33 (64.7)	18 (35.3)	51 (48.1)	
high school or higher	33 (62.3)	20 (37.7)	53 (51.9)	
Smoking				0.788
Yes	4 (66.7)	2 (33.3)	6 (5.8)	
No	62 (63.3)	36 (36.7)	98 (94.2)	
Alcohol consumption				0.384
Yes	42 (61.8)	26 (38.2)	68 (64.4)	
No	24 (66.7)	12 (33.3)	36 (34.6)	
Balanced diet				0.105
Yes	4 (40.0)	6 (60.0)	10 (9.6)	
No	62 (66.0)	32 (34.0)	94 (90.4)	
Leisure-time physical activity				0.019
Yes	21 (50.0)	21 (50.0)	42 (40.4)	
No	45 (72.6)	17 (27.4)	62 (59.6)	
Duration of diabetes (years)	15.1 ± 9.9 (1–42)	12.8 ± 11.7 (1–52)	14.2 ± 10.7 (1–52)	0.306
Family history of T2DM				0.395
Yes	47 (66.2)	24 (33.8)	71 (68.3)	
No	19 (57.6)	14 (42.4)	33 (31.7)	
Chronic complications of diabetes				0.009
None	22 (53.7)	19 (46.3)	41 (39.4)	
1	21 (56.8)	16 (43.2)	37 (35.6)	
2 or more	23 (88.5)	3 (11.5)	26 (25.0)	
HbA1c (%)	8.0 ± 1.0 (7.0–11.6)	6.3 ± 0.5 (5.0–6.9)	7.4 ± 1.2 (5.0–11.6)	<0.001
BMI (kg/m^2^)	29.6 ± 5.1 (18.8–40.8)	28.1 ± 6.1 (17.0–43.0)	29.1 ± 5.5 (17.0–43.0)	0.162
WC (cm)	104.4 ± 12.6 (71.0–134.0)	94.9 ± 15.2 (60.0–130.0)	100.9 ± 14.3 (60.0–134.0)	0.001
TG (mg/dL)	150.9 ± 81.4 851.0–507.0)	145.0 ± 77.7 (57.0–377.0)	148.8 ± 79.7 (51.0–107.0)	0.719
HDL-C (mg/dL)	47.2 ± 15.1 (23.0–90.0)	53.8 ± 16.7 (31.0–88.0)	49.6 ± 15.9 (23.0–80.0)	0.042
LDL-C (mg/dL)	94.9 ± 29.2 (47.6–147.0)	92.8 ± 31.9 (22.4–158.2)	94.1 ± 30.1 (22.4–158.2)	0.738
Total cholesterol (mg/dL)	171.8 ± 31.1 (122.0–227.0)	177.2 ± 31.7 (101.0–224.0)	173.7 ± 31.3 (101.0–227.0)	0.399
hs-CPR (mg/L)	2.9 ± 2.3 (0.2–8.5)	1.8 ± 2.3 (0.0–9.9)	2.5 ± 2.3 (0.0–9.9)	0.030
Number of teeth	21.9 ± 4.9 (6–28)	22.7 ± 4.6 (9–28)	22.2 ± 4.8 (6–28)	0.410
Periodontitis				<0.001
No/mild periodontitis	1 (11.1)	8 (88.9)	9 (8.7)	
Moderate periodontitis	13 (46.4)	15 (53.6)	28 (26.9)	
Severe periodontitis	52 (77.6)	15 (22.4)	67 (64.4)	
Full-mouth PISA (mm^2^)	1342.3 ± 487.9 (422.0–2732.0)	946.1 ± 454.1 (229.0–2252.0)	1204.1 ± 507.7 (229.0–2732.0)	<0.001

BMI: Body Mass Index; HbA1c: glycated hemoglobin; WC: waist circumference; TG: triglycerides; HDL-C: High-density-lipoprotein cholesterol; LDL-C: Low-density-lipoprotein cholesterol; hs-CRP: high-sensitivity C-reactive protein; PISA: periodontal inflammation surface area; T2DM: type 2 diabetes.

**Table 2 jcm-10-01787-t002:** Sociodemographic and clinical characteristics of T2DM patients (mean ± SD (range) or *n* (%)) according to the periodontal status.

Variable	No/Mild Periodontitis (*n* = 9)	Moderate Periodontitis (*n* = 28)	Severe Periodontitis (*n* = 67)	Total	*p* Value
Age (years)	60.1 ± 17.1 (40–79)	66.0 ± 9.7 (47–80)	65.8 ± 8.9 (43–80)	65.3 ± 10.1 (40–80)	0.266
Sex					0.240
Males	3 (4.9)	16 (26.2)	42 (68.9)	61 (58.7)	
Females	6 (14.0)	12 (27.9)	25 (58.1)	43 (41.3)	
Education level					0.902
<high school	5 (9.8)	14 (27.5)	32 (62.7)	51 (48.1)	
high school or higher	4 (7.5)	14 (26.4)	35 (66.0)	53 (51.9)	
Smoking					0.696
Yes	1 (16.7)	1 (16.7)	4 (66.6)	6 (5.8)	
No	8 (8.2)	27 (27.5)	63 (64.3)	98 (94.2)	
Alcohol consumption					0.455
Yes	5 (7.3)	15 (22.1)	48 (70.6)	68 (65.4)	
No	4 (11.1)	13 (36.1)	19 (52.8)	36 (34.6)	
Balanced diet					0.196
Yes	0	1 (10.0)	9 (90.0)	10 (9.6)	
No	9 (9.6)	27 (28.7)	58 (61.7)	94 (90.4)	
Leisure-time physical activity					0.904
Yes	4 (9.5)	12 (28.6)	26 (61.9)	42 (40.4)	
No	5 (8.1)	16 (25.8)	41 (66.1)	62 (59.6)	
Duration of diabetes (yrs)	14.9 ± 11.2 (1–38)	12.9 ± 11.9 (1–52)	14.7 ± 10.1 (1–42)	14.2 ± 10.7 (1–52)	0.754
Family history of T2DM					0.015
Yes	7 (9.9)	13 (18.3)	51 (71.8)	71 (68.3)	
No	2 (6.1)	15 (45.4)	16 (48.5)	33 (31.7)	
Chronic complications of diabetes					0.002
None	5 (12.2)	12 (29.3)	24 (58.5)	41 (39.4)	
1	4 (10.8)	15 (40.5)	18 (48.6)	37 (35.6)	
2 or more	0 (0.0)	1 (3.8)	25 (96.2)	26 (25.0)	
Glycemic control					<0.001
Good	8 (21.1)	15 (39.5)	15 (39.5)	38 (36.5)	
Poor	1 (1.5)	13 (19.7)	52 (78.8)	66 (63.5)	
HbA1c (%)	6.1 ± 0.7 (5.0–7.3)	7.1 ± 1.3 (5.4–10.0)	7.6 ± 1.1 (5.2–11.6)	7.4 ± 1.2 (5.0 –11.6)	0.001
BMI (kg/m^2^)	31.0 ± 6.3 (23.0–43.0)	27.9 ± 6.0 (18.0– 40.3)	29.3 ± 5.1 (17.0–40.8)	29.1 ± 5.5 (17.0–43.0)	0.295
WC (cm)	100.2 ± 11.3 (80.0–115.0)	98.4± 16.9 (66.0–130.0)	102.0 ± 13.5 (60.0M–134.0)	100.9 ± 14.3 (60.0–134.0)	0.533
TG (mg/dL)	136.1 ± 92.5 (57.0–374.0)	176.1 ± 109.4 (63.0–107.0)	139.0 ± 59.6 (51.0–358.0)	148.8 ± 79.7 (51.0–107.0)	0.103
HDL-C (mg/dL)	52.2 ± 17.4 (38.0–83.0)	47.1 ± 15.6 (23.0–83.0)	50.3 ± 16.0 (24.0–90.0)	49.6 ± 15.9 (23.0–80.0)	0.592
LDL-C (mg/dL)	94.8 ± 43.6 (22.4–154.0)	91.9 ± 31.9 (38.4 –158.2)	95.0 ± 27.7 (47.6 –147.0)	94.1 ± 30.1 (22.4–158.2)	0.903
Total cholesterol (mg/dL)	179.1 ± 29.0 (124.0–217.0)	174.6 ± 35.5 (101.0–207.0)	172.6 ± 30.0 (122.0–226.0)	173.7 ± 31.3 (101.0–227.0)	0.832
hs-CPR (mg/L)	0.7 ± 0.06 (0.0–2.0)	1.5 ± 1.2 (0.0–4.7)	3.1 ± 2.5 (0.2–9.9)	2.5 ± 2.3 (0.0–9.9)	<0.001
Number of teeth	24.1 ± 3.8 (18–28)	22.1± 4.8 (10–28)	22.0 ± 4.9 (6–28)	22.2 ± 4.8 (6–28)	0.472
Full-mouth PISA (mm^2^)	415.5 ± 105.5 (229.0 – 575.0)	859.4 ± 199.5 (494.0 –1298.0)	1454.1 ± 431.3 (585.0 – 2732.0)	1204.1 ± 507.7 (229.0 –2732.0)	< 0.001

BMI: Body Mass Index; HbA1c: Glycated hemoglobin; WC: Waist circumference; TG: Triglycerides; HDL-C: High-density-lipoprotein cholesterol; LDL-C: Low-density-lipoprotein cholesterol; hs-CRP: High-sensitivity C-reactive protein; PISA: Periodontal Inflamed Surface Area; T2DM: type 2 diabetes.

**Table 3 jcm-10-01787-t003:** Association between severe periodontitis and glycemic control.

Model and Variables	Severe Periodontitis (Dichotomous)		
	OR	95% IC	*p* Value
Model 1			
Glycemic control (poor vs. good)	4.574	1.724 to 12.138	0.002
Family history of T2DM (yes vs. no)	3.323	1.189 to 9.283	0.022
hs-CRP (mg/L)	1.656	1.187 to 2.306	0.003
Model 2			
HbA1c (%)	1.608	1.034 to 2.502	0.035
Family history of T2DM (yes vs. no)	3.257	1.194 to 8.885	0.021
hs-CRP (mg/L)	1.692	1.213 to 2.360	0.002
	**PISA (mm^2^)**		
	**ß**	**95% IC**	***p* Value**
Model 3			
Glycemic control (poor vs. good)	297.419	104.887 to 489.951	0.003
Chronic diabetes complications (at least one vs. none)	205.264	19.895 to 390.632	0.030
hs-CRP (mg/L)	46.002	6.136 to 85.868	0.024
Model 4			
HbA1c (%)	89.601	11.265 to 167.937	0.025
Chronic diabetes complications (at least one vs. none)	219.628	30.917 to 408.339	0.023
hs-CRP (mg/L)	49.949	9.412 to 90.486	0.016

HbA1c: Glycated hemoglobin; hs-CRP: High-sensitivity C-reactive protein; PISA: Periodontal Inflames Surface Area; OR: Odds ratio; 95% IC: 95% interval confidence; β: Unstandardized coefficient.

**Table 4 jcm-10-01787-t004:** Association between poorly controlled T2DM and severe periodontitis.

Model and Variables	Poorly Controlled T2DM (Dichotomous)		
	OR	95% IC	*p* Value
Model 5			
Severe periodontitis (yes vs. no)	8.509	2.988 to 24.230	<0.001
Leisure-time physical activity (yes vs. no)	0.384	0.143 to 1.033	0.058
Balanced diet (yes vs. no)	0.125	0.027 to 0.580	0.008
WC (cm)	1.057	1.018 to 1.096	0.004
Model 6			
PISA (mm^2^)	1.002	1.001 to 1.003	<0.001
Leisure-time physical activity (yes vs. no)	0.397	0.150 to 1.051	0.063
Balanced diet (yes vs. no)	0.140	0.027 to 0.725	0.019
WC (cm)	1.062	1.023 to 1.103	0.002

PISA: Periodontal Inflamed Surface Area; WC: Waist circumference; OR: Odds ratio; 95% IC: 95% interval confidence.

## Data Availability

The data presented in this study are available on request from the corresponding author.

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
