# Peer review of "Bidirectional Association between Metabolic Control in Type-2 Diabetes Mellitus and Periodontitis Inflammatory Burden: A Cross-Sectional Study in an Italian Population"

_jcm, 2021, doi:10.3390/jcm10081787_

Round 1

Reviewer 1 Report

The authors conducted a study that aimed to determine the periodontal health status of 104 Type-2 Diabetes mellitus patients attending an Outpatient Diabetes Center in North Italy.

Periodontitis was assessed by CDC-AAP-classification and inflammatory burden by PISA index. Additional variables were sociodemographic and life style factors, laboratory tests including glycated hemoglobin. The authors concluded that there is a strong bidirectional connection between periodontitis and poor glycemic control.

This study is well written and understandable. Although the topic was intensively studied in the past, the results are of scientific interest. The strength of this study lies in the number of relevant diabetes and especially periodontitis parameters that show both, periodontitis extend (CDC-AAP), and inflammatory burden (PISA).

My major concern is the emphasis of the term; bi-directional association. This technical term was initially developed in behavioral science and statistics. There are also some studies that found bidirectional associations with T2DM and pancreatitis (Lee et al. 2016; doi: 10.1097/MD.0000000000002448) and depression (Golden et al 2009; doi: 10.1001/jama.299.23.2751). Both studies in a longitudinal design, with much higher sample sizes. I recommend to elaborate if and why your statistical approach is valid to find this strong bidirectional association that you wrote in the conclusions.

Minor concerns regarding the statement of adhering to the STROBE checklist (line 83): No supplementary STROBE-list is provided for reviewing, and some items, i.e., #9:bias and #10:sample size, are not reported.

Lines 111-112: What do you mean with: "Repeated measurements of periodontal parameters were performed in 20% of the sample.” Is this for intra-examiner calibration? Where the initial measurements erroneous and had to be corrected?

Lines 217-218: What is the math behind this figure? Which variables define the grey-space. Did you account for any interdependencies? I feel that a more detailed explanation is needed, how you reached those results you present in figure1.

Author Response

Dear Editor,

Thank you for the comments regarding our manuscript (jcm-1164805) entitled “Bidirectional association between metabolic control in type-2 diabetes mellitus and periodontitis inflammatory burden: a cross-sectional study in an Italian population” by F. Romano et al.

The Reviewers comments spurred us to further improve the overall quality of our message.

Yours faithfully,

Giovanni N Berta

Authors’ responses to Reviewers:

Reviewer #1: The authors conducted a study that aimed to determine the periodontal health status of 104 Type-2 Diabetes mellitus patients attending an Outpatient Diabetes Center in North Italy.

Periodontitis was assessed by CDC-AAP-classification and inflammatory burden by PISA index. Additional variables were sociodemographic and life style factors, laboratory tests including glycated hemoglobin. The authors concluded that there is a strong bidirectional connection between periodontitis and poor glycemic control.

This study is well written and understandable. Although the topic was intensively studied in the past, the results are of scientific interest. The strength of this study lies in the number of relevant diabetes and especially periodontitis parameters that show both, periodontitis extend (CDC-AAP), and inflammatory burden (PISA).

Comment 1 to the Authors: My major concern is the emphasis of the term bi-directional association. This technical term was initially developed in behavioral science and statistics. There are also some studies that found bidirectional associations with T2DM and pancreatitis (Lee et al. 2016;doi:10.1097/MD.0000000000002448) and depression (Golden et al 2009; doi: 10.1001/jama.299.23.2751). Both studies in a longitudinal design, with much higher sample sizes. I recommend to elaborate if and why your statistical approach is valid to find this strong bidirectional association that you wrote in the conclusions.

Authors’ response/action: We thank the reviewer for his helpful comments. We used along the manuscript the term “bidirectional” because there is a long established evidence for an independent and two-way association between T2DM and periodontitis (Taylor et al. 2001). In the periodontitis-DM direction severe periodontitis is associated with poor glycemic control and most severe DM complications (Borgnakke et al. 2013), while in the DM-periodontitis direction hyperglycemia is associated with more severe periodontal breakdown and poorer response after non-surgical and surgical periodontal treatment. This bidirectional association has been recently emphasized in the Joint Workshop on Periodontal Diseases and Diabetes by the International Diabetes Federation and the European Federation of Periodontology (Sanz et al. 2018).

Thus, we have attenuated the emphasis and we have removed the term “bidirectional” from the aim and the conclusions of the study to be more consistent.

Comment 2 to the Authors: Minor concerns regarding the statement of adhering to the STROBE checklist (line 83): No supplementary STROBE-list is provided for reviewing, and some items, i.e., #9:bias and #10:sample size, are not reported.

Authors’ response/action: Please, enclose you can find STROBE-list as requested. We have acknowledged the small sample size and the convenience sampling between the limitations of the current study (lines 370 – 373).

Comment 3 to the Authors: Lines 111-112: What do you mean with: "Repeated measurements of periodontal parameters were performed in 20% of the sample.” Is this for intra-examiner calibration? Where the initial measurements erroneous and had to be corrected?

Authors’ response/action: Measurements of periodontal parameters were repeated in 20% of the original sample by the clinician involved in the study and compared with those recorded by an experienced examiner who served as “gold reference”. This was performed in order to assess the intra- and inter-examiner reproducibility. We obtained levels of agreement comparable with those previously reported in the literature, thus we considered in the analysis only the first recorded measurements without any correction. We have improved the text in order to be clear (lines 117 – 120).

Comment 4 to the Authors: Lines 217-218: What is the math behind this figure? Which variables define the grey-space. Did you account for any interdependencies? I feel that a more detailed explanation is needed, how you reached those results you present in figure1.

Authors’ response/action: Thank you for your advice. We have added more details in the legend to make the figure clearer. We have explained what the grey-area represents and also added a ruler.

STROBE Statement—Checklist of items that should be included in reports of cross-sectional studies

Item No

Recommendation

Title and abstract

1

(a) Indicate the study’s design with a commonly used term in the title or the abstract Page 1

(b) Provide in the abstract an informative and balanced summary of what was done and what was found Page 1

Introduction

Background/rationale

2

Explain the scientific background and rationale for the investigation being reported Page 2

Objectives

3

State specific objectives, including any prespecified hypotheses Page 2

Methods

Study design

4

Present key elements of study design early in the paper Page 2

Setting

5

Describe the setting, locations, and relevant dates, including periods of recruitment, exposure, follow-up, and data collection Page 2

Participants

6

(a) Give the eligibility criteria, and the sources and methods of selection of participants Pages 2-3

Variables

7

Clearly define all outcomes, exposures, predictors, potential confounders, and effect modifiers. Give diagnostic criteria, if applicable Page 3

Data sources/ measurement

8*

For each variable of interest, give sources of data and details of methods of assessment (measurement). Describe comparability of assessment methods if there is more than one group Page 3

Bias

9

Describe any efforts to address potential sources of bias Page 10

Study size

10

Explain how the study size was arrived at Page 10

Quantitative variables

11

Explain how quantitative variables were handled in the analyses. If applicable, describe which groupings were chosen and why Page 4

Statistical methods

12

(a) Describe all statistical methods, including those used to control for confounding Pages 3-4

(b) Describe any methods used to examine subgroups and interactions Pages 3-4

(c) Explain how missing data were addressed NA

(d) If applicable, describe analytical methods taking account of sampling strategy NA

(e) Describe any sensitivity analyses NA

Results

Participants

13*

(a) Report numbers of individuals at each stage of study—eg numbers potentially eligible, examined for eligibility, confirmed eligible, included in the study, completing follow-up, and analysed Figure 1

(b) Give reasons for non-participation at each stage Figure 1

(c) Consider use of a flow diagram Figure 1

Descriptive data

14*

(a) Give characteristics of study participants (eg demographic, clinical, social) and information on exposures and potential confounders Pages 4-5

(b) Indicate number of participants with missing data for each variable of interest

Outcome data

15*

Report numbers of outcome events or summary measures Pages 5-6

Main results

16

(a) Give unadjusted estimates and, if applicable, confounder-adjusted estimates and their precision (eg, 95% confidence interval). Make clear which confounders were adjusted for and why they were included Pages 7-8

(b) Report category boundaries when continuous variables were categorized Pages 5-6

(c) If relevant, consider translating estimates of relative risk into absolute risk for a meaningful time period NA

Other analyses

17

Report other analyses done—eg analyses of subgroups and interactions, and sensitivity analyses NA

Discussion

Key results

18

Summarise key results with reference to study objectives Pages 8-9

Limitations

19

Discuss limitations of the study, taking into account sources of potential bias or imprecision. Discuss both direction and magnitude of any potential bias Page 10

Interpretation

20

Give a cautious overall interpretation of results considering objectives, limitations, multiplicity of analyses, results from similar studies, and other relevant evidence Pages 9-10

Generalisability

21

Discuss the generalisability (external validity) of the study results Page 10

Other information

Funding

22

Give the source of funding and the role of the funders for the present study and, if applicable, for the original study on which the present article is based Page 11

*Give information separately for exposed and unexposed groups.

Reviewer 2 Report

  • The current version of the manuscript seems like a long outline of a PhD thesis with too many details. Although a great number of significant data was collected, the way of the manuscript wrote may decrease the potential value of the study, and also enthusiasm of the readers.

  • As the authors mentioned previously the use of PISA in the association between DM and PD has been previously investigated in the literature (1111/j.1600-051X.2009.01377.x) Therefore authors should clarify the clinical novelty of the current study. Please also revise the conclusion section based on the current findings of the study.

  • Do you think that the presence of negative or positive control groups (periodontitis+systemically healthy or DM+ non-periodontitis) may increase the potential value of the current study?

  • Discussion section has a fluidity problem with a great number of excessive paragraphs. For instance, several paragraphs mentioned the role of CRP. It should be outlined in a single paragraph. Likewise, the role inflammatory mediators should be summarized. Overall, a revision of this section may increase the enthusiasm of the readers.

  • The association between PD and other types of DM (T1DM) should also be mentioned in the introduction. (DOI: 10.1016/j.archoralbio.2017.06.009)

  • Abstract should be revised and reorganized. Line 22-23 there is no need to define PISA in the abstract, the number of patients and also their age range should be included.

  • Line 48. Periodontitis should be defined with a sentence in the first use.

  • How can clinicians use PISA in the clinical practise particularly in DM patients?

  • Were there any correlation between CAL PD and PISA?

  • Were T2DM patients under IV injections or oral medications?

  • Please mention the limitations of the current study.

  • Till the results section no information regarding the collection of serum CRP levels.

  • Please simplify the results section to reduce confusion and improve enthusiasm. Too many information on the tables may cause confusion...

Author Response

Reviewer #2:

Comment 1 to the Authors: As the authors mentioned previously the use of PISA in the association between DM and PD has been previously investigated in the literature (1111/j.1600-051X.2009.01377.x). Therefore, authors should clarify the clinical novelty of the current study. Please also revise the conclusion section based on the current findings of the study.

Authors’ response/action: The study by Nesse et al. (2009) demonstrated the presence of a dose-response relationship between HbA1c levels and PISA values in T2DM patients. In the present study we explored the association between periodontitis and poor glycaemic control in both periodontitis-T2DM and T2DM-periodontitis direction. In addition, we quantified the periodontal damage in terms of amount of inflamed and ulcerated epithelial tissue (PISA Index) and severity of periodontitis based on clinical CDC/AAP case definition. We have considered this aspect along the revised discussion (e.g. lines 267 – 272).

Comment 2 to the Authors: Do you think that the presence of negative or positive control groups (periodontitis+systemically healthy or DM+ non-periodontitis) may increase the potential value of the current study?

Authors’ response/action: We thank the reviewer for his helpful suggestion. We carried out a cross-sectional study among outpatients of a Diabetic Center in North Italy in order to assess their periodontal conditions and to relate them to the degree of metabolic control. Interestingly, the majority of the study population we selected suffered from moderate to severe periodontitis, only a minority had periodontally healthy conditions. This is a clinically relevant finding because there are vey few data in Italy on this topic. The presence of positive and negative control groups may further strengthen the two-way association between periodontitis and T2DM. However, this requires a different study design. It will be interesting to analyze these aspects in a future study.

Comment 3 to the Authors: Discussion section has a fluidity problem with a great number of excessive paragraphs. For instance, several paragraphs mentioned the role of CRP. It should be outlined in a single paragraph. Likewise, the role inflammatory mediators should be summarized. Overall, a revision of this section may increase the enthusiasm of the readers.

Authors’ response/action: Thank you for your suggestion. We have reorganized and revised the Discussion section entirely.

Comment 4 to the Authors: The association between PD and other types of DM (T1DM) should also be mentioned in the introduction. (DOI: 10.1016/j.archoralbio.2017.06.009)

Authors’ response/action: We are grateful with the Reviewer to arise this point. We have mentioned in the Introduction the association between periodontitis and T1DM (lines 38 – 41 and lines 48 – 49).

Comment 5 to the Authors: Abstract should be revised and reorganized. Line 22-23 there is no need to define PISA in the abstract, the number of patients and also their age range should be included.

Authors’ response/action: Thank for your useful comment. We have revised the abstract and added demographic information as suggested.

Comment 6 to the Authors: Line 48. Periodontitis should be defined with a sentence in the first use.

Authors’ response/action: We have added a definition of periodontitis in the Introduction Section (lines 49 – 52).

Comment 7 to the Authors: How can clinicians use PISA in the clinical practice particularly in DM patients?

Authors’ response/action: PISA quantifies the amount of inflamed periodontal tissue, thus it numerically represents the active inflammatory status of the periodontium. Accordingly, PISA is an index that has been used in the periomedicine research to study the association, if any, between the periodontal inflammatory burden and presence/severity of systemic diseases. Its calculation is based on complex tooth-specific algorithms that take into consideration clinical attachment level (CAL), probing depth (PD) and bleeding on probing (BoP). Beside being not practical to use PISA in the clinical practice, PISA is a measure of the degree of periodontal inflammation and not allows to perform a clinical diagnosis.

The periodontal diagnosis made in the clinical practice is a summation of the information from the medical and dental histories, combined with the findings of the clinical examination. The actual clinical classification of periodontal diseases and conditions (Tonetti et al. 2018) is based on full-mouth recording of clinical parameters (presence/absence of bacterial plaque, presence/absence of BoP, PD, CAL, tooth mobility, furcation involvement) at six sites per tooth.

Comment 8 to the Authors: Were there any correlation between CAL PD and PISA?

Authors’ response/action: PISA is strictly correlated with CAL and PD values as they are utilized in the formula proposed by Hujoel et al (2001) and Nesse (2009) for its calculation.

Comment 9 to the Authors: Were T2DM patients under IV injections or oral medications?

Authors’ response/action: Thank you for your comment. We have added a sentence from line 201 to line 204.

Comment 10 to the Authors: Please mention the limitations of the current study.

Authors’ response/action: We have extended the paragraph on the limitations of the study as suggested (lines 358 –373).

Comment 11 to the Authors: Till the results section no information regarding the collection of serum CRP levels.

Authors’ response/action: We have added in the Introduction section a sentence on the role of circulating pro-inflammatory mediators as CRP in T2DM patients (lines 60 – 62).

Comment 12 to the Authors: Please simplify the results section to reduce confusion and improve enthusiasm. Too many information on the tables may cause confusion...

Authors’ response/action: Thank you for the suggestion. We have reorganized and simplified Tables 1 and 2 as well as the Results Section.

Reviewer 3 Report

In general, the manuscript is written clearly and an interesting topic. I only have some minor comments below:

  1. A CONSORT flowchart is advised to add in the manuscript to show the number of patients included in each stage.
  2. The authors used multivariate linear or logistic regression analysis to assess the association between poorly T2DM and several predictors (Table 4) and between perio and several predictors (Table 3). The authors are advised to use backward selection in the regression analysis to exclude the predictors which are not significant in the multivariate analysis.
  3. In the method section, the authors collected multiple variables such as diet, lesure-time physical, etc. The authors are advised to briefly explain why those variables were included as the potential predictors in the study? It is based on the previous literature, or clinical perspective, etc?
  4. in the statistical analysis section, the authors are advised to pre-define which statistical test was used to assess the univariate association between categorical predictors (e.g. sex) and the categorical outcomes (poor/good glymic control, or mild/moderate/severe periodontitis). I guess it is chi-square test, but the authors did not mention this in the statistical analysis section.
  5. How did the authors correct the significance level with Bonferroni correction?

Author Response

Dear Editor,

Thank you for the comments regarding our manuscript (jcm-1164805) entitled “Bidirectional association between metabolic control in type-2 diabetes mellitus and periodontitis inflammatory burden: a cross-sectional study in an Italian population” by F. Romano et al.

The Reviewers comments spurred us to further improve the overall quality of our message.

Yours faithfully,

Giovanni N Berta

Authors’ responses to Reviewers:

Reviewer #3: In general, the manuscript is written clearly and an interesting topic. I only have some minor comments below:

Comment 1 to the Authors: A CONSORT flowchart is advised to add in the manuscript to show the number of patients included in each stage.

Authors’ response/action: We have added a flowchart as suggested (Fig1).

Comment 2 to the Authors: The authors used multivariate linear or logistic regression analysis to assess the association between poorly T2DM and several predictors (Table 4) and between perio and several predictors (Table 3). The authors are advised to use backward selection in the regression analysis to exclude the predictors which are not significant in the multivariate analysis.

Authors’ response/action: Thank you for your helpful advice. We used a stepwise backward selection method in the regression analysis to identify predictors of poor glycaemic control and periodontitis severity. The significance of the contribution of the variables to the model was estimated and compared with the removal criterion (p < 0.1). When a potential predictor met the removal criterion, it was removed from the regression model. The model was then re-estimated for the remaining predictor variables, and the process was repeated until no further predictors met the removal criterion. Thus, we retained the waist circumference in the models because the corresponding coefficients are close to the statistically significance and there are many published articles demonstrating the statistical association between WC and metabolic control.

Comment 3 to the Authors: In the method section, the authors collected multiple variables such as diet, leisure-time physical, etc. The authors are advised to briefly explain why those variables were included as the potential predictors in the study? It is based on the previous literature, or clinical perspective, etc.

Authors’ response/action: We collected diet, environmental and life-style variables based on the available evidence in the literature of their role in increasing the risk of T2DM and in modulating the metabolic control (Kolb and Martin 2017). We have added a sentence in the material and method section to clarify this aspect (lines 169 – 171).

Comment 4 to the Authors: in the statistical analysis section, the authors are advised to pre-define which statistical test was used to assess the univariate association between categorical predictors (e.g. sex) and the categorical outcomes (poor/good glycaemic control, or mild/moderate/severe periodontitis). I guess it is chi-square test, but the authors did not mention this in the statistical analysis section.

Authors’ response/action: We apologize for the missing information. We used the chi-square test. We have added it in the statistical analysis section (lines 152 – 153).

Comment 5 to the Authors: How did the authors correct the significance level with Bonferroni correction?

Authors’ response/action: We used the Kruskal-Wallis test and post-hoc test for pair-wise comparisons. As multiple tests are being carried out, SPSS makes an adjustment to the p-value. The Bonferroni adjustment is to multiply each Dunn’s p-value by the total number of tests being carried out.

Round 2

Reviewer 2 Report

I would like to thank the authors for addressing all the concerns that I have had regarding the manuscript. The only minor change that I would like to suggest is that some clinicians may desire to use PISA in the clinical practise. Therefore, to prevent any misunderstanding a single sentence regarding the impracticality of PISA during routine clinical practise in the discussion section of the manuscript should increase the potential value of the manuscript. 

Author Response

Comment 1 to the Authors: I would like to thank the authors for addressing all the concerns that I have had regarding the manuscript. The only minor change that I would like to suggest is that some clinicians may desire to use PISA in the clinical practice. Therefore, to prevent any misunderstanding a single sentence regarding the impracticality of PISA during routine clinical practise in the discussion section of the manuscript should increase the potential value of the manuscript.

Authors’ response/action: Thank you for your useful comment. We have added a sentence to clarify this aspect in the Discussion section (lines 282-285).

This manuscript is a resubmission of an earlier submission. The following is a list of the peer review reports and author responses from that submission.